# Drivers to Obesity—A Study of the Association between Time Spent Commuting Daily and Obesity in the Nepean Blue Mountains Area

**DOI:** 10.3390/ijerph19010410

**Published:** 2021-12-31

**Authors:** Ivan Parise, Penelope Abbott, Steven Trankle

**Affiliations:** 1Department of General Practice, School of Medicine, Western Sydney University, Sydney, NSW 2751, Australia; I.Parise@westernsydney.edu.au; 2School of Medicine, Western Sydney University, Sydney, NSW 2751, Australia; S.Trankle@westernsydney.edu.au

**Keywords:** obesity, overweight, non-communicable disease, health promotion, built environment, urban environment, sprawl, public transport, commute time

## Abstract

Obesity has become a public health challenge in every country on this planet, with a substantial contribution to global mortality and morbidity. Studies of the built environment have shown some promise in understanding the drivers of this obesity pandemic. This paper contributes to this knowledge, by focusing on one aspect of the urban environment and asking whether there is an association between commuting and obesity in residents of the Nepean Blue Mountains area on the fringes of Sydney. This is a cross-sectional study with obesity being the dependent variable, and commuting the independent variable, where 45 min or less was defined as local and distant commute was more than 45 min. In the sample of 158 respondents, the risk of obesity was twice as likely in the distant commuters than in the local commuters (OR 2.04, 95% CI 1.051 to 3.962, *p =* 0.034). Investigation of possible mediators of this association was limited by sample size; however, mode of transport was found to be a significant mediator. The results support the design of cities to provide health supporting environments for all residents, including equitable access to employment at a reasonable distance and effective public transport.

## 1. Introduction

“Health is created and lived by people within the settings of their everyday life; where they learn, work, play, and love.”—The Ottawa Charter, 1986.

By the end of the twentieth century, for the first time in history, the human population had more overweight than underweight people [1]. In the short historical period of industrialisation, more efficient food production and increased access to calorically dense food, (largely in the form of vegetable oils and sugars), caused increased stature and then increasing body mass index (BMI) (weight in kilograms divided by height in metres squared) [2]. The associated increased burden of disease attributable to non-communicable diseases (NCD) is not just restricted to high income countries, high BMI (BMI greater than 25) is now a problem in every country [3].

Obesity (BMI ≥ 30) is a major public health challenge [4]. Australia has one of the greatest prevalence of high BMI in developed countries [5]: in 2018, an estimated 67% of the Australian population had a high BMI, while over 31% were obese [6]. In 2019, high BMI was estimated to be responsible for nearly 11% of deaths and over 8% of the total disability adjusted life years (DALYs) in Australia [7]. High BMI was the top risk factor for disability in 2019, with smoking ranked second [7]. These figures are explicable given obesity’s association with the modern epidemics of NCDs such as diabetes mellitus, cardiovascular disease, osteoarthritis, renal disease, colon cancer and depression [8]. In twenty years, global mortality associated with high BMI has increased by over 55%, while DALYs have increased by nearly 90% [7].

Given the severity of the obesity pandemic, there is an urgent need to understand the drivers behind it. Since the early part of this century, the built environment has attracted much interest as a contributing factor to obesity [9,10,11,12,13,14,15]. Currently, just over half the world’s population live in cities and it is estimated to reach two-thirds by 2050, making the urban environment (UE) the most important built environment human populations are exposed to [16]. Consequently, development of the UE impacts most humans on this planet—setting the context for livelihood, recreation, and health behaviour.

A city’s development depends on many drivers such as historical, geographical, demographic, and economic, which vary in different regions of the world. Australia has distinct influences that have shaped the growth of its UE [17]. Access to freshwater in a continent that is 35% desert means that most large cities are situated on the coastal or hinterland zones. Unlike older regions such as the United Kingdom and Europe, Australian cities were developed largely after industrialisation and especially under the influence of motor vehicles. In the second half of the 20th century, a desire for single residences on a “1/4 acre” block (about 1000 sq metres), increased prosperity from resources, increasing population largely through immigration, reduced cost of cars, and increased government spending on roads led to cities with very low density by world standards [17]. This has created sprawling cities such as Sydney. Sydney houses about 5.3 million people [18], roughly 65% of the state of NSW’s population, and given the above-mentioned drivers, has expanded to cover a total area of around 12,300 square kilometres [19].

Sydney’s development is a consummate example of urban sprawl, which consists of rapid expansion geographically, low density housing, separate single purpose land use, consequent long distances between different purposes such as residence and employment, and increased reliance on motor vehicles. Urban sprawl has been found to be associated with obesity, the evidence gathering steadily over the last twenty years, with a greater prevalence of obesity found on the fringes of sprawling cities [13,20,21,22,23,24,25]. Different reasons for the association have been posited, with some studies providing evidence that long commute times created by sprawl interfere with the ability to maintain healthy behaviours [26,27,28,29,30,31]. One explanation is that being time-poor creates a compromise between commuting and preparing meals, exercising, and other activities [30]. The over reliance on motor vehicles that exists in urban sprawl has also been postulated to mediate the increased rates of obesity found [32,33,34,35,36,37,38].

Obesity rates are not only higher in sprawling cities, but the distribution is not homogeneous throughout the city. Areas on the fringes have a higher incidence than those closer to the centre of a city [25,39]. The population studied in this research reside on the fringes of the sprawling city of Sydney, with residences between 60 and up to 120 km from the central business district. The area is categorised as the Nepean Blue Mountains Primary Health Network area (NBMPHN) and is defined in more detail in the sample section of the method. In 2014–2015, this area had the highest incidence of obesity in urban primary health network areas of Australia [40], and in 2019, the age standardised proportion of high BMI in NBMPHN was 63.8% compared to 58.9% in Northern Sydney [41]. The NBMPHN area demonstrates the characteristics of urban sprawl, with nearly all people living in low density housing with major areas of employment long distances away, and nearly 70% of workers using a car in 2016 for commuting [42].

It is not clear what factors contribute to the observed increase in obesity on the fringes of sprawling cities. This study addresses this by investigating a specific parameter of the built environment, the time taken to commute, to see if it is associated with a higher risk of obesity. Possible mediators of any association found, such as diet, exercise, sleep, stress, and mode of transport are also investigated. The information collected by this study will help to inform development and construction of cities that equitably distributes environments to encourage healthy behaviour and decrease chronic conditions such as obesity.

## 2. Methods

### 2.1. Ethical Approval

Ethical approval for this project has been granted by the Western Sydney University Human Research Ethics Committee in accordance with the National Statement on Ethical Conduct in Human Research 2007 (Updated 2018), approval number: H13558.

### 2.2. Research Question

The primary research question was: is there an association between commuting to a distant place, (greater than 45 min), of full time and obesity in residents of NBMPHN area? Secondarily, this study investigated the contribution of mediators known to be associated with obesity such as diet, exercise, sleep, stress, and mode of transport.

### 2.3. Sample and Data Collection

The population of this cross-sectional study reside on the western fringes of Sydney, the sample being drawn from the Nepean Blue Mountains Primary Health Network (NBMPHN) area. A primary health network area is an administrative health region created to deliver access to primary care services for residents of that area, of which there are 31 in Australia. The NBMHN comprises the local government areas of Penrith, Blue Mountains, Lithgow and Hawkesbury; however, the sample was drawn from patients attending three general practices in the Blue Mountains and Nepean. These two government areas are typically suburban, with Penrith being a larger city centre of over 212,000 people residing in low to medium density housing, while the Blue Mountains, with a population of over 79,000 people, live in low density houses spread through small towns on the radial ridges of a World Heritage national park.

General Practitioners from the practices were invited to participate and provided their informed consent. Patients were invited to participate if they were: employed full-time, as per Australian Bureau of Statistics (ABS) definition of working 35 h or more a week; a resident of the NBMPHN area as defined by the local government health areas; and working at a site of employment that was unchanged for two years prior to the commencement of the study. This period was chosen to provide enough time for the employment to have influenced health behaviour and the development of obesity. It is difficult to know exactly how long an aspect of the environment may take to have an impact on a population and the development of obesity; however, Eid et al. employed a 2–4-year period after a change in neighbourhood environment to assess change in obesity rates [43]. Ewing et al. also used data over a 2–4-year period in their study of urban sprawl and obesity [20]. Handy et al. used a one year period to ascertain changes in activity levels [44].

Patients were excluded from the study if they had chronic conditions that had prevented them from pursuing their normal exercise and eating patterns, were off work due to illness for a continuous period of more than 6 weeks in the last year, were pregnant, suffered a heart attack or stroke in the last 2 years, or had had surgery for obesity.

If the patient agreed to participate, the BMI was measured by their doctor and written on the information sheet for the respondent to insert in the questionnaire. The questionnaire was accessed electronically offsite via an anonymous link and had no identifiers. There was no way for the doctor to know if a patient completed the questionnaire. A small number of written questionnaires were also supplied and were stored in a sealed box at the surgeries. The electronic data were stored on secure servers at Western Sydney University, transferred via a cloud link, while the few written questionnaires were scanned and transferred to the servers.

### 2.4. Dependent Variable

Obesity, the dependent variable, was deemed present if a BMI greater than or equal to 30 was recorded, and absent if BMI was less than 30.

### 2.5. Independent Variable

The independent variable is the time taken to commute to work in a typical working week, the possible responses being less than 15 min; 15–45 min, 46–90 min; or greater than 90 min. These categories are based on a previous study of commuting and well-being [45]. The responses were then categorised into a dichotomous variable of local and distant, with local commute defined as travel time up to and including 45 min, and distant as more than 45 min. This categorisation is consistent with other research investigating the effect of travel on health indicators [45,46].

### 2.6. Confounders

Possible confounders of age, gender, level of education, household income, marital status were recorded [47,48], having access to exercise facilities at work [49], and attitude to diet and physical activity [50,51,52,53]. Age was recorded in years, and gender as male, female, or other. Level of education was categorised as high school or less, trade certificate/diploma, and University. Household income was categorised based on the ABS 2016 census [54], and divided into five groups with average weekly gross incomes in dollars of: less than AUD $650, $650 to $1249, $1250 to $1999, $2000 to $2999, $3000 or more. Marital status was categorised as single or married/in de facto relationship.

Attitudes to healthy behaviour have been shown to influence actual behaviour [55]. Attitude to healthy eating was assessed using a 5-point Likert scale with items ranging from ‘strongly agree’ to ‘strongly disagree’ based on surveys of parental attitudes and effect on pre-schooler nutrition [56,57]. Similarly, attitudes to physical activity were briefly assessed using the same Likert scale.

### 2.7. Mediators

Variables that have been shown to be associated with obesity are physical activity, eating behaviour, sleep, stress, and mode of transport. Longer commutes are associated with having less time to allocate to other activities and affect the above mentioned factors [30,49,50,58,59], while also impacting choice of mode of transport [46]. Therefore, measures of these variables were included to ascertain mediation of any association between commuting and obesity.

Physical activity has been shown to affect many health indicators, apart from its role in the development of obesity [60]. Consequently, the average number of hours spent in moderate intensity physical activity (walking, housework, leisure activity) during a typical working week was asked, with the following possible responses: less than 1 h, 1 h to less than 2 ½ h, 2 ½ h or more. To capture those who choose to partake in vigorous exercise (gym, running, cycling, etc.), the average number of the hours spent in vigorous activity was asked with the possible responses: none, less than 1 ¼ h, and 1 ¼ h or more. Activity was then categorised into adequate or inadequate according to World Health Organisation guidelines in adults (18–64 years) [60]; adequate moderate intensity activity was ≥2 ½ h, while adequate vigorous activity was 75 min or more.

Capturing caloric intake is more problematic, especially given self-reporting biases. However, authors have used the number of meals purchased away from home as an indicator of high caloric intake. One study estimated that one additional meal purchased away from home a day increased daily caloric intake by about 134 calories, with the quality of food intake deteriorating [61]. Another, associated two meals or more a week with an increased risk of obesity [62]. Consequently, this study divided the number of meals bought away from home into less than two, two to five, and more than five. The responses were dichotomised and deemed likely to promote obesity if respondents bought more than five meals away from home a week. Beverages are also important caloric inputs. Despite much debate, the consensus is that increased consumption of high sugar drinks is associated with obesity [63,64,65,66]. One study found that one or more standard cans of soft drink a day was associated with obesity [64]. Accordingly, the number of cans or glasses of soft drinks consumed in an average working week was asked in the categories of none, less than seven and seven or more per week. Obesogenic response was deemed to be drinking seven or more cans/glasses of sugary drinks per week.

Alcohol intake is a contributor to caloric intake [62]. The respondent was also asked to record average number of standard alcoholic drinks (100 mL glass of wine, 285 mL of full-strength beer, 60 mL port or sherry, 30 mL/nip of spirits) per week as either none, ten or less standards a week, and more than ten (according to Australian guidelines).

Inadequate sleep has been associated with obesity [67], and excessive work hours has been associated with a trade-off between exercise, social time, and sleep [68]. The average hours of sleep a night in a usual working week was asked with the possible responses of less than seven, seven to eight, and more than eight hours. Healthy sleep was deemed seven to eight hours a night.

Stress is both a possible consequence of the tension between differing elements in one’s life and commuting [26,69], and has been directly associated with obesity [70]. Understanding stress and capturing it is an involved procedure. Due to limitations of the questionnaire, a single question based on a panel of questions by Cohen et al. [71] asked how often the respondent had felt stressed or unable to cope with what they had to do in the past year, with the possible answers: never, rarely, sometimes, fairly often and very often.

Finally, mode of transport has previously been associated with obesity and level of physical activity [20,72], and length of commute has been shown to affect mode of transport [46]. Consequently, principal mode of transport to work was asked and responses categorised into private vehicle (car, motorbike), public transport (train, bus), or active transport (walking, bicycle).

### 2.8. Statistical Analysis

The study data were analysed using IBM SPSS Statistics version 27. The rates of obesity were calculated for those who travel to employment outside the local area and those travelling to employment in the local area. The chi square statistic was used to test the null hypothesis that the rate of obesity is independent of whether the resident travels to a local place of employment or to a distant site of employment.

The initial estimate of sample size to enable analysis of confounders by logistic regression was 226. This was calculated using the rule of 10 cases per variable, the 8 confounding variables and a known rate of obesity of 35.5% [40]. However, the COVID pandemic and consequent decreased presentation to doctor’s surgeries, limited the sample collection dramatically. The smaller sample size of 158 took much longer to achieve than was expected. Consequently, we compared the local commute group to the distant, to determine whether they were matched with respect to the confounders. With continuous numerical variables, an independent t test was performed; with dichotomous categorical variables, a chi square test was performed; and with ordinal categorical variables, a Mann–Whitney U test was performed. The Likert scales were considered ordinal as there were only five categories [73].

The significance of possible mediators, previously described, were analysed using Hayes’ PROCESS v 5.5.

## 3. Results

Over a 14-month period, 160 respondents were recruited, with 158 having full data for analysis. In the sample of 158, 95 people reported that they worked locally and 63 people commuted 45 min or more. The rate of obesity in residents working locally was 29.5%, whereas the rate in those commuting more than 45 min was 46.0%. The overall rate of obesity in the respondents was 36.0% which is almost the same as found in the NBMPH are by the Australian Institute of Health and Wellness (35.5%) [40]. The risk of being obese in residents commuting 45 min or more was twice that of residents working locally (OR 2.04, 95% CI 1.051 to 3.962, *p* 0.034).

### 3.1. Sample Characteristics

The sample consisted of nearly 40% male, 60% female and one respondent that identified as other. In total, 94% of respondents resided in the Blue Mountains, with the other 6% in the Nepean area. About three quarters were married or in a de-facto relationship, 60% were university trained while less than 8% had education to high school or less. Weekly income among the respondents was found to be below $1250 in 18.8%, between $1250 and $2999 in 56.4%, and $3000 and over in 24.8%. Stress was said to be present never or sometimes in over 63%, over 95% strongly agreed/or somewhat agreed that eating was important to health, and over 96% agreed strongly/somewhat agreed that exercise was important to health.

The confounding variables were shown to be evenly distributed between both groups (Table 1), neutralising their impact. The one confounder that was close to being significant was marital status; however, there is an association between being married and more obese [47] and therefore this would bias in favour of increased obesity in the local group.

### 3.2. Analysis of Possible Mediators to the Risk of Obesity

Overall, in the respondents, moderate exercise was found to be adequate in about 60%; intense exercise was adequate in about 31%; number of bought meals were obesogenic in about 32%; there was a high intake of sugary drinks in 6.6%; only 50% slept for the recommended hours; less than 8% drank more than the recommended amount of alcohol; and 74% drove a motor vehicle to work. A breakdown according to commute of the possible mediators for the effect of commuting on the risk of obesity is shown below (Table 2).

Obesity is not directly caused by time spent commuting, it is caused by the net positive caloric intake—calories consumed versus calories spent. Based on this, the above variables were investigated as possible mediators by which longer commute could be associated with obesity. Unfortunately, this study was of too low power to examine the mediation effects comprehensively. However, by using logistic regression and an arbitrary p cut off 0.25, five variables were identified to be investigated. This was possible given the assumption of 10 active cases required per variable investigated, and the frequency of obesity in this study being 36% required a sample size of at least 138. Using Hayes’ PROCESS v 5.5, the following relationships were determined in the pathway between commuting and obesity (Figure 1). The only significant pathway was that mediated by mode of transport. The effect was shown to be that with more active transport the risk for obesity decreased by 26% (−0.26, 95% CI −0.6689, −0.0169).

Private vehicle usage was much higher in the local commuters (70% of all private car users), while the use of public transport was much higher in the distant commuters (93% of all public transport users). Active transport was largely limited to the local commuters 8%.

Stratification of the relationship of obesity to local and distant commuters by mode of transport revealed further trends (see Table 3). Like other studies that find higher rates of obesity for longer times spent in a car [35,36,38], the distant commuters who used a private vehicle had a rate of obesity of nearly 56% compared with 34% in the local commuters. Public transport, on the other hand, which involves serendipitous exercise, demonstrated an obesity rate of 31% in the distant commuters.

## 4. Discussion

The primary finding of this study was that people residing in the Nepean Blue Mountains who travel more than 45 min to a place of employment have twice the risk of obesity compared to those who work locally. These results contribute to understanding the drivers behind the observed higher rates of obesity in sprawling cities [20,24], especially the increased prevalence the further one lives from a city centre [25]. The investigation of the association between sprawling cities and obesity is still in its early stages. Commuting time has been proposed as a risk, though there are few studies investigating the relationship between commute and obesity. Frank et al. presented an association between land use mix and car use with obesity [13]. Another study found an association between longer commute and directly measured BMI; however, the study sample came from different suburban types, making it difficult to rule out the impact of other built environment parameters [74]. Zhang et al., using self-reported BMI, found an association between obesity and neighbourhoods with greater reliance on automobiles, though not in large fringe metropolitan neighbourhoods [75]. Another study disputed the importance of distance from the centre of a city, suggesting that the local neighbourhood parameters such as walkability were more important [76].

Despite being a small study, the primary finding of this research strengthens the evidence that longer commutes are associated with obesity. The sample population is drawn largely from one urban neighbourhood type and thus diminishes the confounding effect of diverse neighbourhood parameters presented in previous studies on commuting and obesity.

By focusing exclusively, for the first time, on a population on the fringe of a sprawling city, the results of this study help to understand the observed higher incidence of obesity on city fringes, particularly in Sydney [39]. Many considerations contribute to the movement of people to the fringes of a city; however, housing affordability is a major factor in choosing where to live [77]. Urban fringe developments are often portrayed as being ‘more affordable’ [78] and median house prices trend lower the further they are from the central business district [79]. This contributes to the finding that disadvantage is often located on the fringe of cities [17]. Given that disadvantaged communities are already burdened by higher prevalence of chronic health risks such as obesity, the major finding of this study amplifies the calls from other researchers to reduce the impact on health by developing cities with an equitable distribution of distance between employment and residence [46,78,80].

In contradistinction to the above demographics of urban fringes, the population sampled in this study had largely middle to high income—nearly 30% were in the upper quintile while over 50% were in the next two quintiles. Nearly 60% of the population studied had university education, whereas a common pattern is that mostly ‘lower-skilled’ workers choose to live on the fringes [78]. This could be due to the distribution of respondents, as 94% resided in the Blue Mountains which is established in a World Heritage national park. Possibly, lifestyle choices contributed to the decision to live in this area. Notwithstanding the higher education and income, there was still a higher risk of obesity in those who had the longer commute.

This study also examined the possible mediators of the association found between long commute and obesity. Despite the smaller sample size limiting the analysis of the possible mediators, mode of transport was found to be a significant mediator. People commuting by private vehicle not only had a higher rate of obesity (41%) compared to those using public transport (31%), but also those commuting by private vehicle to distant sites had a much higher rate of obesity (56%) than those driving locally (34%). These results add to the evidence supporting the importance of public transport to mitigate obesity in populations [37,81,82,83], and further reinforce the need to decrease commute time, as the longer one is in a car daily, the greater the risk of obesity [13].

Unfortunately, public transport on the fringes of large cities like Sydney is commonly deficient and only increases the already greater reliance on private motor vehicles [84]. Additionally, transport services are inequitably distributed across populations according to advantage, with the wealthiest benefitting the most [85]. Given the mediation effects of mode of transport on obesity rates found in this study, public transport must be strengthened in outer urban areas like the Nepean Blue Mountains, not just for the liveability of neighbourhoods, but also for the health of populations living there.

The built environment is more than the mere context for health, it is an integral part of the health of populations—a health resource. Health is a universal right. If this is to be upheld, there needs to be equitable access to health resources which includes environmental assets. If we ignore this and allow cities to become sprawling and have centres of employment far from residences, with poor transport systems, we will only be creating a built environment that inequitably distributes chronic illnesses such as obesity.

### Strengths and Limitations

This is a cross-sectional study, and hence no causality is claimed. It is also limited by the smaller sample size, which affected the power of the study and the ability to determine the contribution of mediators. Recruitment was much less than anticipated, largely because of the COVID-19 pandemic that saw about a 50% reduction in presentations to General Practice clinics. A further study with a larger sample size and power would help to investigate the contribution of exercise and diet to the observed higher risk of obesity in people commuting to distant locations.

The mediating effects of mode of transport may be subject to reverse causality, that is that people who are obese choose to travel by car rather than undergo the serendipitous exercise that occurs when travelling by public transport. There are, however, many other studies that would refute the possibility of reverse causality with mode of transport, including an experimental study that shows a change in weight when mode of transport is altered [37,81,82,83,86]. Moreover, reverse causality cannot explain the effect of a longer car commute on the risk of obesity. The prevalence of obesity is higher in the distant car commuter group (56%) compared to the local car commuter group (34%), which is a finding supported by other research [13,35,36].

The smaller number of distant commuters may represent the difficulty presented to this population finding time to visit a doctor. It is hard to determine the bias of recruiting from a population presenting to a general practice. It could mean that there was a bias towards those who were more unwell, although equally it could bias to those who are more interested in health prevention. However, as stated in the results, the overall rate of obesity in the respondents was 36.0% which was almost the same as that already reported in the NBMPH area (35.5%), thus making bias less likely.

Strengths of this study included that the weight of the respondent was measured by an independent observer. Self-reported height and weight have been regarded as problematic in previous surveys, especially given associated recall and social desirability bias [37,46,87], yet only a minority of studies of the effect of sprawling cities on obesity measured BMI directly [88]. Another strength is that the population was drawn from one locality, with similar neighbourhoods which diminishes the effect of self-selection bias [89], and the critique that people with a tendency to obesity choose to live in sprawling suburbs [43]. People in this study chose to live in a low-density neighbourhood on the fringes of a large city, they exhibited similar attitudes to the importance of diet and exercise to health, yet there was a clear difference between residents with a long commute versus those who worked locally. The study of a specific urban area is a strength from an analysis point of view; however, it limits the results from being generalized to other urban neighbourhoods, for example inner urban commuters.

## 5. Conclusions

The findings in this paper reiterate the importance of drivers to obesity outside the direct control of the individual, providing evidence that commuting to a distant workplace increases the risk of obesity. It is of vital importance that governments, regulators, planners, and developers understand the impacts of the design of our cities on the health of residents. This paper supports the need for the design of cities that incorporates equitable access to employment at a reasonable distance, and the fair provision of efficient, affordable public transport services.

## Figures and Tables

**Figure 1 ijerph-19-00410-f001:**
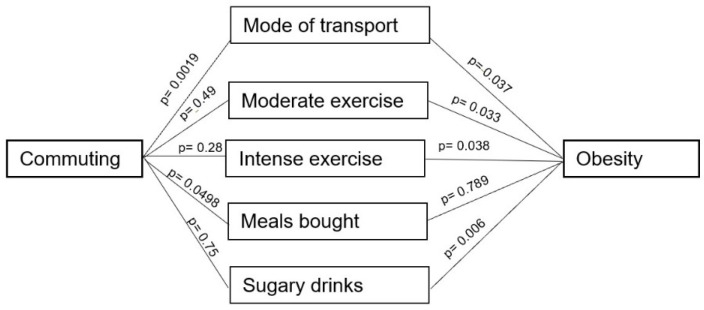
Possible mediators in the association between obesity and commuting.

**Table 1 ijerph-19-00410-t001:** Confounding variables to the association between obesity and commuting.

Variable		Local	Distant	*p*
Age				0.912
	Mean	50.25	50.48	
	Std Dev.	12.50	13.40	
Gender				0.72
	Male	36 (37.9%)	24 (37.8%)	
	Female	58 (61.1%)	39 (61.9%)	
	Other	1 (1%)	0	
Marital status	Married/in de facto relationship/Single/divorced/widowed	75 (79.8%)	42 (66.7%)	0.064
		19 (20.2%)	21 (33.3%)	
Education				0.67
	High school or less	9 (9%)	4 (6.1%)	
	trade/certificate/diploma	32 (32%)	21 (31.8%)	
	University undergraduate/postgraduate	59 (59%)	41 (62.1%)	
Income				0.089
	Less than $650	3 (3%)	3 (4.6%)	
	$650 to $1249	19 (19%)	6 (9.2%)	
	$1250 to $1999	28(28%)	15 (23%)	
	$2000 to $2999	28 (28%)	22 (33.8%)	
	$3000 or more	22 (22%)	19 (29.2%)	
Exercise at Work				0.249
	Yes	13 (13.7%)	13 (20.6%)	
	No	82 (86.3%)	50 (79.4%)	
Attitude to eating				0.114
“Is Healthy Eating (e.g., decreasing sweets/pastries, increasing fibre and vegetables) important to health?”	Strongly agree	74 (74%)	39 (59.1%)	
	Somewhat agree	22 (22%)	23 (34.8%)	
	Neither agree nor disagree	1 (1%)	4 (6.1%)	
	Somewhat disagree	2 (2%)	0	
	Strongly disagree	1 (1%)	0	
Attitude to exercise				0.347
“Exercise is important to my health/wellbeing?”	Strongly agree	77 (77%)	44 (66.7%)	
	Somewhat agree	19 (19%)	20 (30.3%)	
	Neither agree nor disagree	1 (1%)	2 (3%)	
	Somewhat disagree	3 (3%)	0	
	Strongly disagree	0	0	

**Table 2 ijerph-19-00410-t002:** Possible mediators.

Variable	Response	Local	Distant
Adequacy of Moderate Exercise	Inadequate	39 (39%)	27 (41%)
	Adequate	61 (61%)	39 (59%)
Adequacy of Intense exercise	Inadequate	72 (72%)	42 (64%)
	Adequate	28 (28%)	24 (36%)
Meals bought	Not Obesogenic	74 (74%)	39 (59%)
	Obesogenic	26 (26%)	27 (41%)
Sugar Drinks	Low Intake	92 (92%)	63 (95%)
	High Intake	8 (8%)	3 (5%)
Sleep hours	Unhealthy	42 (42%)	41 (62%)
	Healthy	58 (58%)	25 (38%)
Weekly Alcohol	At or below Guidelines	94 (94%)	59 (89%)
	Exceeding Guidelines	6 (6%)	7 (11%)
Mode of transport	Private vehicle (car/motorbike)	86 (89%)	35 (53%)
	Public transport (train/bus)	2 (2%)	29 (44%)
	Bicycle or walk	9 (9%)	2 (3%)
Stress	Never	12 (12%)	9 (14%)
	Sometimes	58 (58%)	26 (39%)
	About half the time	14 (14%)	17 (26%)
	Most of the time	11 (11%)	14(21%)
	Always	5 (5%)	0

**Table 3 ijerph-19-00410-t003:** Stratification of mode of transport according to work locality.

Mode of Transport	Work Locality		No Obesity	Obesity	Total
Private vehicle (car/motorbike)	Local	Count	54	28	82
		% Within Work Locality	65.90%	34.10%	100.00%
	Distant	Count	15	19	34
		% Within Work Locality	44.10%	55.90%	100.00%
Public transport (train/bus)	Local	Count	2	0	2
		% Within Work Locality	100.00%	0.00%	100.00%
	Distant	Count	18	9	27
		% Within Work Locality	69.00%	31.00%	100.00%
Bicycle or walk	Local	Count	8	0	8
		% Within Work Locality	100.00%	0.00%	100.00%
	Distant	Count	1	1	2
		% Within Work Locality	50.00%	50.00%	100.00%

## Data Availability

Publicly archived dataset analyzed or generated during the study, available in SPSS format https://cloudstor.aarnet.edu.au/plus/s/ExvjHtXmtQYz67b (accessed on 27 April 2021).

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
