# Peer review of "Drivers to Obesity—A Study of the Association between Time Spent Commuting Daily and Obesity in the Nepean Blue Mountains Area"

_ijerph, 2021, doi:10.3390/ijerph19010410_

Round 1

Reviewer 1 Report

This is a very well written paper investigating the impact of travel times and obesity among adult workers living in the Nepean Blue Mountains area on the fringes of Sydney.  The authors report that those who travelled more than 45 minutes had double the odds of obesity compared to those who travelled shorter times.  While limited by sample size (n=158), there is a suggestion that the association may be mediated by mode of transport - in particular, those who travelled >45 mins by car had far greater rates of obesity (55.9%) than those who travelled >45 mins by public transport (31.0%).  The authors conclude that "The results support the design of cities to provide health supporting environments for all residents, including equitable access to employment at a reasonable distance and effective public transport."

I really enjoyed this paper, which contained interesting insights despite the small sample size. I have just minor comments which may improve the paper.

  1. The authors acknowledge that there may be biases from recruiting from patients attending GP clinics and posit that it is unclear how these biases might operate. As such, would it be possible to compare the achieved sample against some descriptives from the population from which the sample was drawn (ideally, adult workers living in the Nepean Blue Mountains area) to get some sense of whether (and if so how) the sample might be biased?
  2. In the limitations section it should be acknowledged that the design is cross-sectional so it is hard to draw conclusions about causality. Related to this, the possibility of reverse causation should be acknowledged.  For example, might those who are obese choose to travel by car rather than by public transport, perhaps because use of public transport typically involves greater physical activity than transport by car?
  3. While the focus on one geographic area is highlighted as a strength as it “diminishes the effect of self-selection bias”, this should also be acknowledged as a limitation to generalisability beyond the area.

There are a few other minor typographical issues to address:

  1. Lines 263-5: “This section may be divided by subheadings. It should provide a concise and precise description of the experimental results, their interpretation, as well as the experimental conclusions that can be drawn”. Are these instructions that haven’t been removed?
  2. Table 2. Check the sleep hours figures for the ‘Distant’ group – these don’t add to 100.  Also, the ‘Distant’ column reports percentages to 1 decimal place whereas the ‘Local’ column reports percentages to 1 decimal place.  Make this consistent.
  3. Line 310. I think ‘found and association’ should be ‘found an association’.

Author Response

Thank you for reviewing my paper, and I am grateful for your input. The answers to your specific questions follow:

  1. The authors acknowledge that there may be biases from recruiting from patients attending GP clinics and posit that it is unclear how these biases might operate. As such, would it be possible to compare the achieved sample against some descriptives from the population from which the sample was drawn (ideally, adult workers living in the Nepean Blue Mountains area) to get some sense of whether (and if so how) the sample might be biased?

Unfortunately, there is no data that I can access that stratifies obesity by the category of worker. However, it is a good point you raise, and one that is answered in the paper. In results I point out that the overall obesity rate in the sample is very similar to that found in the area according to the Australian Institute of Health and Welfare data. I have included this in the discussion in the redraft.

  1. In the limitations section it should be acknowledged that the design is cross-sectional so it is hard to draw conclusions about causality. Related to this, the possibility of reverse causation should be acknowledged.  For example, might those who are obese choose to travel by car rather than by public transport, perhaps because use of public transport typically involves greater physical activity than transport by car?

As I was only positing association and not causation, I inferred lack of evidence for causation. I have remedied this in the re-draft and stated it explicitly.

Reverse causation should be acknowledged, however the investigation of mode of transport was meant to investigate a possible mediator to the effect of a long commute on obesity. There are many other studies that investigate this specifically, including experimental studies that show a change in weight when mode of transport is altered. However, I agree that I could state this, and have added this to the re-draft.

Reverse causality could explain the difference between private vehicle and active transport; however, it cannot explain the effect of long car commute on risk of obesity. The prevalence of obesity is higher in the long commute compared to the local car commute. 

  1. While the focus on one geographic area is highlighted as a strength as it “diminishes the effect of self-selection bias”, this should also be acknowledged as a limitation to generalisability beyond the area.

The study does not attempt to generalise to other neighbourhood types. There have been other studies showing this effect across different neighbourhood types. I have, however, stated this limitation in the re-draft.

  1. Lines 263-5: “This section may be divided by subheadings. It should provide a concise and precise description of the experimental results, their interpretation, as well as the experimental conclusions that can be drawn”. Are these instructions that haven’t been removed?

Yes, they are left over from the template, will remedy this….. they were instructions in the template, left in by accident in the transfer of my manuscript, and have been deleted in the re-draft.

  1. Table 2. Check the sleep hours figures for the ‘Distant’ group – these don’t add to 100.  Also, the ‘Distant’ column reports percentages to 1 decimal place whereas the ‘Local’ column reports percentages to 1 decimal place.  Make this consistent.

The number was a typographical error, decimal points also made uniform.

  1. Line 310. I think ‘found and association’ should be ‘found an association’.

Altered in re-draft.

Reviewer 2 Report

This is an interesting study highlighting commuting time as a risk factor for obesity, despite the small sample size. The small sample size indeed is the limitation of the current study. However, this data is still publishable given the appropriate analysis.

The authors may consider the following to improve this manuscript

'Obesity has become one of the major public health challenges in every country on this planet' is this claim true ? There are still countries living with insufficient food supply.

In the first paragraph of discussion, can the authors provide rationale why commuting time is a risk factor for obesity?

Please also highlight novelty of this study in the discussion. This is a finding in Australia, are the authors expecting similar pattern in other part of the world? Please comment and discuss.

Author Response

Thank you for reviewing my paper, and I am grateful for your input. The answers to your specific questions follow:

'Obesity has become one of the major public health challenges in every country on this planet' is this claim true? There are still countries living with insufficient food supply.

The statement as it stands, over emphasises the challenge in some countries, however every country is facing problems created by obesity in some sector of their population. Countries living with insufficient food now have the double burden of disease attributable to both under and over nutrition. Additionally, the rapid growth of obesity in low and especially middle-income countries is concerning. This is supported by the reference in the article and other reviews (The 2016 Global Nutrition Report, https://www.who.int/news-room/fact-sheets/detail/obesity-and-overweight)

In the re-draft I have modified the sentence in the abstract to say: “obesity has become a public health challenge in every…”.

In the first paragraph of discussion, can the authors provide rationale why commuting time is a risk factor for obesity?

Although it is convention to place the result in the first paragraph, I thought the flow of the discussion was best served by the way the paper is written. However, I have altered the discussion to follow convention, and placed the reason in the first paragraph.

Please also highlight novelty of this study in the discussion. This is a finding in Australia, are the authors expecting similar pattern in other part of the world? Please comment and discuss.

This area of study is new, however there are many other studies that look at the built environment and obesity, including the studies quoted on the effect of commuting on obesity. I have, however, adapted the discussion in the re-draft to highlight this.

The choice of a sample from one neighbourhood, on the fringes of a sprawling city, attempts to remove the effect of other elements of the neighbourhood on the outcomes. Such elements of the built environment as provision of adequate green space, walkability, access to affordable whole food versus access to high density food outlets are different in different neighbourhoods. This is to diminish the confounding effects if commuters were exposed to different environments, which is a problem in some studies that sample across different neighbourhood types. Given that both those who commute locally and those who commute distantly live in the same neighbourhood, with the same neighbourhood environment, this study suggests that it is the distant commute that is associated with the increased risk of obesity. This is likely to apply to other parts of the world, as it is not the neighbourhood characteristics, but the commuting that is associated with the obesity.

Other evidence supporting this is presented in the paper, such as the prevalence of obesity is higher on the fringes of sprawling cities in other parts of the world. Ewing’s paper in 2003 found that people living in sprawling suburbs, on the fringes of large cities in the U.S. had a higher rate of obesity[1]. Sugiyama et al showed that the abdominal girth is greater the further from the centre of the city [2]. Although this is in Adelaide, another Australian city, the urban structure is very different to Sydney. Additionally, Zhang et al found that the greater dependence on private motor vehicles in the U.S. was associated with a higher risk of obesity[3]. This was replicated in Spain by Nunez et al [4], and in Canada[5]. The greater dependence on private vehicles for commuting is not only characteristic of the area we studied, but of many different sprawling cities around the world.

  1. Ewing R, Schmid T, Killingsworth R, Zlot A, Raudenbush S. Relationship between Urban Sprawl and Physical Activity, Obesity, and Morbidity. American Journal of Health Promotion. 2003;18(1):47-57. doi:10.4278/0890-1171-18.1.47
  2. Sugiyama T, Niyonsenga T, Howard NJ, Coffee NT, Paquet C, Taylor AW, et al. Residential proximity to urban centres, local-area walkability and change in waist circumference among Australian adults. Preventive Medicine. 2016;93:39-45. doi:10.1016/j.ypmed.2016.09.028
  3. Zhang X, Holt JB, Lu H, Onufrak S, Yang J, French SP, et al. Neighborhood commuting environment and obesity in the United States: An urban–rural stratified multilevel analysis. Preventive Medicine. 2014;59(1):31-6. doi:10.1016/j.ypmed.2013.11.004
  4. Núñez-Córdoba JM, Bes-Rastrollo M, Pollack KM, Seguí-Gómez M, Beunza JJ, Sayón-Orea C, et al. Annual Motor Vehicle Travel Distance and Incident Obesity: A Prospective Cohort Study. American Journal of Preventive Medicine. 2012;44(3). doi:10.1016/j.amepre.2012.10.019
  5. Swanson KC, McCormack GR. The relations between driving behavior, physical activity and weight status among Canadian adults. Journal of physical activity & health. 2012;9(3):352. doi:10.1123/jpah.9.3.352

Reviewer 3 Report

The sample size is too small (158 respondents), which prevents us from gaining a reliable and enriched understanding of this issue. I suggest sampling more people to reach a safer conclusion.

Author Response

Thank you for reading my paper, however I have to challenge your rejection of the research, and your statement that follows.

The sample size is too small (158 respondents), which prevents us from gaining a reliable and enriched understanding of this issue. I suggest sampling more people to reach a safer conclusion.

The sample size of 158 was lower and took longer to achieve than we had expected. This, as explained in the paper, was largely due to the COVID pandemic dramatically decreasing presentation of patients to the general practice sites where enrolment occurred. During this difficult time in primary care, most of the consultations were done via teleconferencing thus preventing direct measurement of BMI.

It is not possible to extend the study for many reasons. It took nearly two years to enrol respondents, we had already asked much of the general practitioners participating in this study, and it would be a large impost on them to extend the study given they are currently stressed by doing their usual work as well as taking on a massive immunisation schedule for COVID. Additionally, there is the theoretical risk that given people have now been working from home for over a year, the findings will be altered by a change in commuting practice.

The sample size was large enough to obtain both a statistically significant result and a 95% confidence interval that does not include an odds ratio of one. The major deficit of the sample size has been a low power, if anything this would miss an association. This makes the odds ratio of a doubled risk, and anything up to nearly 4 times the risk of obesity in people with a distant commute, even more important knowledge to communicate to the scientific world.

I agree that it would have been better to have a richer analysis by exploring the possible mediators, and the low power of this study limited this. Again, given the low power, the capture of the association of mode of transport as a mediator to the effect of commuting on obesity is notable. If anything, this paper could stimulate further research into possible mediators.

Given the above discourse, I think it is important to publish this paper, even with the smaller sample size, as the knowledge contributes to an understanding of how the built environment can impact the health of populations, especially given the burden of disease presented by obesity.

Round 2

Reviewer 3 Report

no further comments.